# FLOW-GUIDED NEURAL OPERATOR FOR SELF-SUPERVISED LEARNING ON TIME SERIES DATA

## ABSTRACT

Self-supervised learning (SSL) is a powerful paradigm for learning from unlabeled time-series data. However, popular methods such as masked autoencoders (MAEs) rely on reconstructing inputs from a fixed, predetermined masking ratio. Instead of this static design, we propose treating the corruption level as a new degree of freedom for representation learning, enhancing flexibility and performance. To achieve this, we introduce the Flow-Guided Neural Operator (FGNO), a novel framework combining operator learning with flow matching for SSL training. FGNO learns mappings in functional spaces by using Short-Time Fourier Transform to unify different time resolutions. We extract a rich hierarchy of features by tapping into different network layers ($l$) and flow times ($s$) that apply varying strengths of noise to the input data. This enables the extraction of versatile representations, from low-level patterns to high-level global features, using a single model adaptable to specific tasks. Unlike prior generative SSL methods that use noisy inputs during inference, we propose using clean inputs for representation extraction while learning representations with noise; this eliminates randomness and boosts accuracy. We evaluate FGNO across three biomedical domains, where it consistently outperforms established baselines. Our method yields up to 35% AUROC gains in neural signal decoding (BrainTreeBank), 16% RMSE reductions in skin temperature prediction (DREAMT), and over 20% improvement in accuracy and macro-F1 on SleepEDF under low-data regimes. These results highlight FGNO's robustness to data scarcity and its superior capacity to learn expressive representations for diverse time-series.

## 1 INTRODUCTION

Time-series data are common across domains such as healthcare (Johnson et al., 2016) and weather forecasting (Pathak et al., 2022). Learning useful supervised representations from temporal signals can be challenging when labels are scarce. Thus, self-supervised learning (SSL) has become a compelling technique, enabling models to exploit large collections of unlabeled time series data. Prior work adapts ideas from natural language processing and computer vision, such as BERT (Devlin et al., 2019), masked autoencoders (MAE) (He et al., 2021), and contrastive objectives (Siméoni et al., 2025). Recently, increasingly capable time-series foundation models (Ansari et al., 2024) and flow-based generative models (Zhang et al., 2025) have gained significant attention. Although they focus on forecasting tasks, their SSL abilities are also of considerable interest.

Despite the progress of self-supervised learning in time-series modeling, learning generalizable representations remains challenging due to the heterogeneous nature of real-world data and the diversity of downstream tasks. Time-series signals are often recorded at different sampling rates, and standardizing them through upsampling or downsampling distorts their intrinsic characteristics. In the DREAMT dataset (Eldele et al., 2024), for instance, wearable device signals are collected at multiple frequencies ranging from 4 Hz to 200 Hz. Aligning such data requires interpolation and resampling steps that risk blurring fine-grained events, such as micro-arousals or transient heart rate variability patterns, thereby contaminating the learned representation space.

In addition to resolution mismatches, downstream tasks often demand representations at different temporal and semantic scales. Sleep-stage classification relies on local patterns in the length of seconds, whereas apnea-hypopnea index (AHI) regression requires integrating information across an entire night. Similar multi-scale demands also arise in other domains: clinical forecasting on MIMIC-III (Johnson et al., 2016) depends on long-term trends in vitals, whereas arrhythmia detection

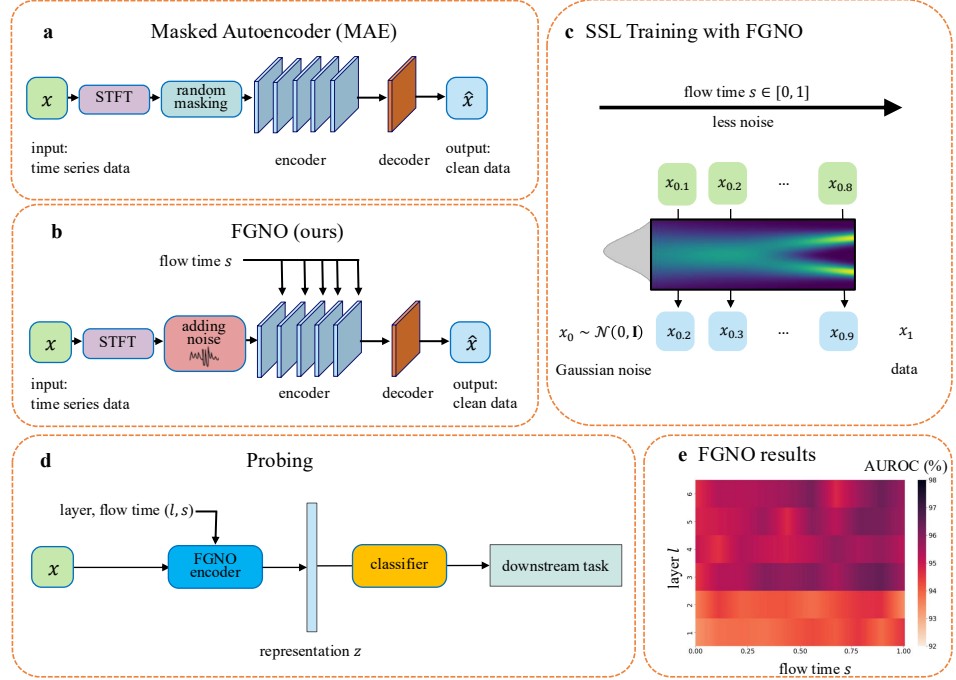

Figure 1: **(a)** A common self-supervised learning (SSL) baseline is Masked Autoencoder (MAE, (He et al., 2021)), where the input data is randomly masked at a *fixed* ratio and then fed to encoders and decoders to reconstruct the clean data. MAE learns useful representations by inpainting the missing part. **(b)** We ask if the ratio can *vary* continuously and propose flow-guided neural operator (FGNO), which is based on flow matching that progressively transforms noisy inputs, corrupted at level $\sigma_s$ (flow time), to clean data by predicting intermediate velocities. Both methods first transform the time series data to spectrograms via STFT (short-time Fourier transform) to extract local time-frequency features. **(c)** FGNO is pre-trained in a self-supervised manner using the flow-matching objective. FGNO learns in function space and empirically shows improved performance across different sampling rates of the input data. The decoder shown is a shallow spectrogram reconstruction head used solely during pretraining and discarded for downstream tasks. **(d)** After SSL pretraining, representations are probed by training a small classifier for downstream tasks. Compared with existing generative SSL methods, we use clean input data instead of noisy data as input and achieve similar performance with no randomness from the noise generation. **(e)** FGNO's performance on sleep/wake classification. A single FGNO model has improved flexibility with various layer and flow time $(l, s)$ combinations.

from electrocardiogram (ECG) relies on millisecond-level waveforms. Despite such needs, most SSL approaches are optimized for a fixed pretraining objective and yield a single latent representation, limiting their adaptability across tasks with different temporal windows.

These considerations motivate *a unified pretraining framework that preserves the fidelity of multi-resolution time-series signals while yielding flexible, task-adjustable representations.*

Neural operators learn mappings directly in the function space of signals, offering a natural framework for time-series modeling where data can be viewed as functions over time (Azizzadenesheli et al., 2024). This approach has achieved state-of-the-art results across various time-series domains, including forecasting, imputation, and anomaly detection (Li & Yang, 2023). Short-Time Fourier Transform (STFT) provides an efficient method in signal processing that focuses on local time-frequency analysis, enabling the extraction of both temporal and spectral features at fine-grained resolutions while being resolution invariant. However, the integration of STFT into operator learning remains largely unexplored, which we try to address in this work.

To adapt to tasks of different abstract levels, generative modeling offers a complementary perspective. Diffusion- and flow-based methods (Ho et al., 2020; Lipman et al., 2023) learn to map simple noise distributions to complex data distributions and are trained with self-supervised signals. Recent studies on images suggest that internal representations taken at different noise levels naturally organize from

low-level textures to high-level global features, providing a *continuous* control knob for multi-scale features (Tang et al., 2023). A concern with these methods, however, is that their inputs are noised at different noise levels, potentially leading to information loss during downstream tasks.

We propose the **Flow-Guided Neural Operator** (FGNO, Fig. 1), a self-supervised framework that pretrains a *flow matching* model on spectrograms of unlabelled data and extracts task-specific representations by selecting a network layer $l$ and flow time $s$ (which we treat interchangeably with noise level). We leverage Short-Time Fourier Transform (STFT) to embed 1D signals into time-frequency representations that preserve both local and global information with resolution invariance. We treat features $\phi_{l,s}(x)$ (the hidden states at layer $l$ conditioned on time $s$ for clean input $x$) as a hierarchy of representations. After pretraining, we train a probing head (classifier) on top of $\phi_{l,s}$ while freezing the backbone. This design turns the flow time $s$ into a practical, tunable degree of freedom that allows users to emphasize fine temporal detail (lower $s$, shallower $l$) or higher-level semantics (higher $s$, deeper $l$) with a *single* pretrained model. Unlike prior generative SSL methods that use noisy inputs during inference, which may contaminate information, FGNO uses clean inputs for representation extraction. This enables superior performance with no randomness.

Empirically, we observe that the optimal choice of network layer $l$ and noise level $s$ is task-dependent: tasks requiring precise local timing benefit from lower noise and earlier layers, whereas tasks relying on global context prefer higher noise and deeper layers. Selecting $(l, s)$ per task yields consistent gains over MAE- and contrastive-based baselines, including up to 35% AUROC improvements on neural signal decoding (BrainTreeBank, Wang et al. (2024a)), 16% RMSE reductions on skin temperature regression (DREAMT, Wang et al. (2024c)). Moreover, our approach demonstrates exceptional robustness to data scarcity: on SleepEDF, FGNO maintains 93.5% accuracy and 89.0% macro-F1 with only 5% labeled data, representing over 20% improvements compared to strong baselines; on Epilepsy, FGNO achieves 94.1% accuracy and 90.3% macro-F1 under the same setting, effectively matching the full-data results.

In summary, our contributions are as follows:

- **An SSL framework combining flow matching and operator learning for time series.** We pretrain flow matching models on time-frequency representations of 1D signals, generated via STFT. This enables training at one resolution while generalizing to other resolutions during downstream tasks, with minimal performance degradation.

- **Flow time as a means to control features.** We demonstrate that flow time $s$ provides an explicit and practical control over representation granularity, yielding a rich, multi-level feature hierarchy by varying the flow time $s$ and network layer $l$.

- **Performance gain with clean input for flow-based representations.** During the probing stage where representations are classified for downstream tasks, we use clean input data instead of noisy data like prior generative SSL methods. We argue that any potential domain gap can be mitigated during probing, and ablation studies show superior performance with no randomness.

- **Empirical advantage on biomedical benchmarks.** FGNO significantly outperforms established baselines across diverse tasks, including up to 35% AUROC improvement in neural signal decoding, 16% RMSE reductions in skin temperature regression, and strong robustness under data scarcity on SleepEDF and Epilepsy—maintaining near full-data performance with only 5% labeled data.

## 2 RELATED WORK

**Self-Supervised Learning (SSL)** SSL has shown great success in learning representations for various downstream tasks without labels, with applications to different modalities such as images (He et al., 2021; Wang et al., 2024b), audio (Gong et al., 2022), and videos (Tong et al., 2022; Gupta et al., 2024). Masked autoencoding (MAE) is a dominant self-supervised paradigm. For instance, BrainBERT (Wang et al., 2023) successfully applies this technique to neural time-series data by training on masked spectrogram representations. Advanced SSL methods for time-series data further enhance representation performance. Contrastive Predictive Coding (CPC, van den Oord et al. (2019)) uses autoregressive predictions and contrastive losses to maximize mutual information in sequential data such as physiological signals. TS-TCC (Eldele et al., 2021) employs temporal-contextual contrasts and augmentations like jittering to improve robustness and generalization. We will compare with these methods in Section 4.

**Generative Models for SSL** Generative models, particularly diffusion and flow matching models, serve as powerful self-supervised learners, as their denoising objective inherently learns rich, multi-level data representations (Fuest et al., 2024). Existing work explores generative models for representation for visual data like images and videos (Fuest et al., 2024; Luo et al., 2023; Vélez et al., 2025; Tang et al., 2023). A dominant technique is to leverage intermediate activations from a pre-trained model's internal layers at various corruption timesteps, creating a feature hierarchy that spans from low-level textures to high-level global features. These extracted features are then used to train lightweight heads for downstream tasks like classification and segmentation. Our work builds on this foundation but shifts the focus from generation to representation learning. We investigate whether the flow matching objective can serve as a powerful self-supervised learning tool for time-series data, enabling the extraction of multi-scale features for downstream discriminative tasks.

**Foundation Models for SSL** Chronos (Ansari et al., 2024) is an autoregressive model based on language model architectures. As a foundation model trained on multiple datasets, Chronos achieves state-of-the-art forecasting results by tokenizing the raw signal and training a GPT-style model. Time-series forecasting has been a heated research area. Ekambaram et al. (2024); Das et al. (2024); Woo et al. (2024) While most SSLs are non-generative, researchers have started to explore how generative models benefit SSLs. (Hemmer & Durstewitz, 2025) Our work primarily focuses on downstream discriminative tasks rather than forecasting.

**Neural Operators** Neural operators (Azizzadenesheli et al., 2024; Kovachki et al., 2023) are deep learning architectures specifically designed to learn mappings between infinite-dimensional function spaces. Neural operators have empirically achieved good performance for approximating the numerical solutions to partial differential equations (PDEs) (Kovachki et al., 2023; Li et al., 2024) and real-world applications such as computational imaging (Jatyani et al., 2025; Wang et al., 2025a;b). A prominent example is the Fourier Neural Operator (FNO, Li et al. (2020)), which leverages the Fast Fourier Transform (FFT) to efficiently model global dependencies. In contrast, our approach utilizes the Short-Time Fourier Transform (STFT), which analyzes signals within local time windows. This allows the model to capture time-varying frequency information effectively and provides greater flexibility in handling time series of different durations.

## 3 METHOD

In this section, we introduce the Flow-Guided Neural Operator (FGNO), a novel framework for self-supervised representation learning. FGNO leverages the flow matching paradigm (Lipman et al., 2023) to learn generalizable representations. By constructing mappings in function space via Fourier-based time-frequency representations (i.e., spectrograms), it functions as a neural operator capable of generalizing across resolutions. The methodology unfolds in two stages: pre-training with flow matching, and probing with representation selection.

### 3.1 SELF-SUPERVISED PRE-TRAINING

**Data Embedding** The pre-training begins with embedding raw 1D signals into a time-frequency representation suitable for functional-space learning. To this end, we apply the Short-Time Fourier Transform (STFT) to convert the input signals $x \in \mathbb{R}^T$ into spectrograms $f \in \mathbb{C}^{N_f \times N_t}$, where $N_f$ denotes the number of frequency bins and $N_t$ the number of time frames. The STFT is defined as

$$f(\tau, \omega) = \int_{-\infty}^{\infty} x(t)w(t - \tau)e^{-j\omega t}dt, \tag{1}$$

with $w(\cdot)$ as a sliding window function (e.g., Hann window) of length $W$, hop size $H$, and $\tau, \omega$ indexing time and frequency. We compute the magnitude spectrogram $\bar{f} = |f|$ as input to our model, which captures both temporal evolution and local patterns. STFT spectrograms are standard for preprocessing in speech recognition and audio analysis (Bäckström et al., 2022) but less common in SSL literature. It is also worth noting that this embedding is resolution-invariant: signals sampled at different rates can be transformed without resampling, avoiding distortions from interpolation.

**Self-Supervised Learning via Flow Matching** Once embedded into magnitude spectrograms, the data functions $\bar{f} \in \mathbb{R}^{N_f \times N_t}$ (sampled from the data distribution $\nu$) are used to pretrain a time-

conditioned neural network $u_\theta(s, g) : [0, 1] \times \mathbb{R}^{N_f \times N_t} \to \mathbb{R}^{N_f \times N_t}$ via flow matching (Kerrigan et al., 2023). Flow matching provides a simulation-free objective for learning continuous normalizing flows that map a simple prior distribution (e.g., Gaussian noise) to the complex data distribution $\nu$. In our self-supervised setup, this objective implicitly learns rich representations by regressing toward a target vector field that guides the denoising of corrupted inputs.

Specifically, for a given timestep $s \sim \mathcal{U}[0, 1]$, we construct a noisy interpolation $g \sim \mu_{\bar{f},s}$ between the clean data function $\bar{f}$ and a noise measure $\pi = \mathcal{N}(0, C_0)$ as

$$g = s\bar{f} + \sigma_s z, \qquad z \sim \pi, \tag{2}$$

where $\sigma_s : [0, 1] \to \mathbb{R}+$ is a monotonically increasing variance schedule that controls the noise level at $s$. The model $u_\theta(s, g)$ is trained to approximate the conditional vector field $v_s^{\bar{f}}(g)$, i.e. the velocity pointing from $g$ toward $\bar{f}$, via the regression loss

$$J(\theta) = \mathbb{E}_{s \sim U[0,1], \bar{f} \sim \nu, g \sim \mu_s^{\bar{f}}} \left[ \left| v_s^{\bar{f}}(g) - u_\theta(s, g) \right|^2 \right], \tag{3}$$

with the target field given by

$$v_s^{\bar{f}}(g) = \frac{(\sigma_s)'}{\sigma_s}(g - s\bar{f}) + \bar{f}, \tag{4}$$

where $\sigma_s'(s)$ denotes the derivative with respect to $s$. Minimizing $\mathcal{J}(\theta)$ equips $u_\theta$ with the ability to simulate the flow ODE

$$\frac{dg}{ds} = u_\theta(s, g) \tag{5}$$

that transports noise to data. This setup ensures self-supervision as the objective solely uses the unlabeled data distribution.

We instantiate $u_\theta$ as a Transformer conditioned on $s$ via sinusoidal positional embeddings concatenated to the input spectrogram channels. The resulting pretrained model encodes multi-scale dynamics—from coarse structures at low $s$ (high corruption) to local details at high $s$ (low corruption)—providing a versatile backbone for downstream probing.

## 3.2 Feature Extraction and Probing

**Feature Extraction with Clean Data**  After pretraining, we freeze the Transformer weights $u_\theta$ and use the model as a feature extractor for downstream tasks. A key challenge here is the distributional shift between the noisy inputs $g$ encountered during pretraining and the clean, labeled samples typically available for fine-tuning. Typical generative SSL approaches address this by generating noisy inputs $g$ at a fixed or sampled timestep $s$ during inference, which introduces randomness and potential information loss from clean downstream data.

To address this while preserving consistency, we extract representations using clean spectrograms $\bar{f}$ as input, conditioning on the timestep $s$ via the pretrained time embeddings, with no explicit noise generation. This deterministic approach yields stable features without randomness or computational overhead. Although training on noisy data and probing on clean data may introduce a domain gap, this is effectively mitigated by the lightweight probing head, which adapts the representations during fine-tuning; our empirical results support this design choice.

Formally, for a clean input spectrogram $\bar{f}$ and desiredflow time $s \in [0, 1]$, the representation at layer $l$ is obtained as

$$\phi_{l,s}(\bar{f}) = u_\theta^{(l)}(s, \bar{f}), \tag{6}$$

where $u_\theta^{(l)}$ denotes the activations at layer $l$ of the frozen model. These $\phi_{l,s}$ form a hierarchy of features: shallower layers and lower $s$ (higher corruption) capture fine details, while deeper layers and higher $s$ emphasize abstract global features.

**Representation Selection and Probing**  The final stage of the FGNO involves training a lightweight probing head (e.g., a linear classifier or regressor) atop selected representations $\phi_{l,s}(\bar{f})$, using labeled data while keeping the backbone $u_\theta$ frozen.

Given that the model captures multiple representations, selecting the optimal representation configuration requires an evaluation of features at various layers $l$ and times $s$. To achieve this, we conduct a grid search over a discrete set of layers and flow times to find the optimal pair $(l^*, s^*)$ that minimizes validation loss:

$$(l^*, s^*) = \arg \min_{l \in L, s \in S} \mathcal{L}_{\text{val}}(l, s). \tag{7}$$

This selection process unlocks FGNO's full potential, allowing users to tailor feature granularity, balancing both temporal and global features. information without retraining the pretrained model.

## 4 EXPERIMENTAL RESULTS

### 4.1 DATASETS AND SETUP

**DREAMT** (Wang et al., 2024c) contains synchronized smartwatch and clinical-grade polysomnography (PSG) data from 100 participants, many with sleep disorders. For FGNO, a single model is pre-trained on the smartwatch's Blood Volume Pulse (BVP) and accelerometer (ACC) signals. This model's features are then evaluated on held-out participants for *two downstream tasks*: a binary sleep/wake classification and a skin temperature regression.

**BrainTree Bank** (Wang et al., 2024a) is a large-scale dataset of intracranial neural responses from 10 subjects watching Hollywood movies (43 hours in total). The dataset includes extensive linguistic annotations of the movie audio, such as transcripts and word onsets. Using a held-out set of subjects for probing, we evaluate our model on a binary speech presence classification task.

**Epileptic Seizure Recognition** (Andrzejak et al., 2001) consists of EEG recordings from 500 subjects, each with 23.6s of brain activity. The original dataset contains five classes, four of which correspond to non-seizure brain states. To focus more on seizure detection, we merge the four non-seizure classes into one and treat this as a binary classification problem (seizure vs. non-seizure).

**SleepEDF** (Goldberger et al., 2000) is a widely used whole-night PSG sleep dataset from PhysioBank. We use a single EEG channel (Fpz–Cz) sampled at 100 Hz and classify each 30-second epoch into five classes: Wake, Non-REM sleep stages N1, N2, N3, and Rapid Eye Movement (REM) sleep.

For all datasets, we follow a strict chronological and subject-based splitting to prevent data leakage. The validation set is kept completely separate from the test set, and all model tuning is performed exclusively on the validation set. Specific preprocessing pipelines, STFT parameters, and detailed data split ratios are provided in Appendix A.

### 4.2 SLEEP CLASSIFICATION AND SKIN TEMPERATURE PREDICTION ON DREAMT

**Comparison to baselines** From Table 1, FGNO significantly outperforms baselines in both sleep classification and skin temperature regression. It yields better AUROC compared to the MAE baseline across the optimal layers. Our peak score (96.4%) also surpasses the gradient boosting approach (92.6%) reported in DREAMT. Notably, our model achieved this using only raw 1D data, whereas the DREAMT baseline required additional clinical metadata (Apnea severity score) (Wang et al., 2024c), highlighting the capability of our self-supervised approach. For skin temperature regression, our best RMSE substantially improves upon both the MAE baseline (0.734°C). Overall, the results highlight FGNO's ability to leverage both network depth and flow time for highly predictive representations, whereas MAE is constrained to layer selection alone.

**Comparison to a foundation model** Time-series foundation models excel in forecasting but may hold potential for SSL. Thus, we benchmark FGNO against Chronos (Ansari et al., 2024), a T5-based family of pretrained models that tokenizes raw signals for autoregressive training. For fairness, we selected a Chronos variant matching FGNO's parameter count and evaluated it as a feature extractor using two strategies: last-token hidden states or average pooling over all states. Average pooling yields Chronos's best results (96.4% AUROC for sleep classification; 0.954°C RMSE for regression), yet FGNO outperforms it narrowly on classification and substantially on regression (37% improvement). This underscores FGNO's role in data-efficient SSL for diverse downstream tasks.

Table 1: Performance comparison on downstream tasks from the DREAMT datasets. Our method (FGNO) achieves superior results on both classification (AUROC) and regression (RMSE) benchmarks compared to baselines.

| Task | Metric | FGNO (Ours) | MAE | Chronos |
|------|--------|-------------|-----|---------|
| Binary Sleep Classification | AUROC (%) ↑ | **96.5** | 95.8 | 96.3 |
| Skin Temperature Regression | RMSE (°C) ↓ | **0.600** | 0.735 | 0.954 |

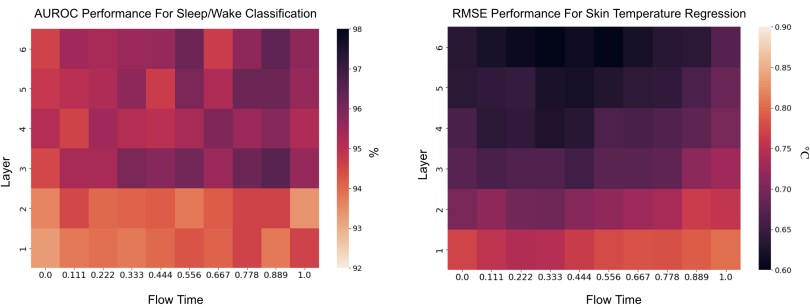

Figure 2: FGNO's performance across different layers and flow times on the DREAMT dataset. **Left:** Sleep classification AUROC (↑). **Right:** Skin temperature regression RMSE (↓). A darker color indicates better performance.

**Insights from different layers and flow times** As shown in Fig. 2, sleep classification accuracy improves substantially in deeper layers. The best AUROC is achieved at layer 3 with low noise ($s = 0.89$). In contrast, skin temperature regression also favors deeper layers but achieves its lowest RMSE at moderate noise levels ($s \in [0.22, 0.56]$). We observe a trend that different levels of abstraction require different flow times. Specifically, the physiological regression task depends more on global features, which align with lower to intermediate flow times (higher corruption levels). Conversely, classification tasks require more local patterns that prefer higher flow times (lower corruption), where temporal features are preserved. This allows us to leverage our understanding of the task to determine whether higher or lower flow times are needed, and reciprocally, to use the flow time to gain deeper insights into the task itself.

## 4.3 CLASSIFICATION TASKS ON BRAINTREEBANK

To further benchmark FGNO against state-of-the-art methods, we compared our model against Brain-BERTWang et al. (2023), PopTChau et al. (2024), and the DeepNN baseline on the BrainTreeBank dataset. As shown in Table 2 , FGNO achieves superior performance across multiples tasks (Speech, Volume, Pitch) despite being significantly smaller (370K parameters vs. 20M+ for baselines). Notably, while PopT incorporates explicit domain knowledge regarding electrode relationships, FGNO outperforms it on 3 out of 4 tasks purely through flow matching representation learning.

Table 2: Model comparison in terms of size and accuracy on all four tasks of BrainTreeBank. The DeepNN's performance is quoted from Chau et al. (2024). FGNO outperforms latest baselines in most tasks while being significantly smaller in size.

| Model | Model Size | Speech | Volume | Pitch | Sentence |
|-------|-----------|--------|--------|-------|----------|
| FGNO (Ours) | 370K | **0.925** | **0.919** | **0.796** | 0.797 |
| BrainBERT | 43M | 0.63 | 0.60 | 0.51 | 0.66 |
| PopT | 20M | 0.93 | 0.87 | 0.74 | **0.90** |
| DeepNN | 20M | 0.70 | 0.64 | 0.56 | 0.71 |

Table 3: Accuracy (ACC) and macro F1-score (MF1) for all models evaluated with both 100% and a scarce 5% of labeled training data. FGNO consistently outperforms other methods, demonstrating its robustness and data efficiency in low-label regimes. **Bold** values indicate the best performance in each column. The baselines' performance is cited from Eldele et al. (2021).

| Baseline | 100% of labeled data | | | | 5% of labeled data | | | |
| | SleepEDF | | Epilepsy | | SleepEDF | | Epilepsy | |
| | ACC | MF1 | ACC | MF1 | ACC | MF1 | ACC | MF1 |
|---|---|---|---|---|---|---|---|---|
| Random Initialization | 35.6 | 23.8 | 90.3 | 81.1 | 22.8 | 22.8 | 75.5 | 70.5 |
| Supervised | 83.4 | 74.8 | 96.7 | 94.5 | 60.5 | 54.8 | 83.4 | 80.4 |
| SSL-ECG (Sarkar & Etemad, 2022) | 74.6 | 65.4 | 93.7 | 95.1 | 73.4 | 65.3 | 92.8 | 89.0 |
| CPC (van den Oord et al., 2019) | 82.8 | 73.9 | 96.4 | 94.4 | 76.3 | 70.5 | 90.2 | 90.2 |
| SimCLR (Chen et al., 2020) | 78.9 | 68.6 | 96.1 | 93.5 | 64.2 | 61.9 | 91.3 | 89.1 |
| TS-TCC (Eldele et al., 2021) | 83.0 | 73.5 | **97.2** | **95.5** | 77.0 | 70.9 | 93.1 | **93.7** |
| **FGNO (ours)** | **93.9** | **89.1** | 94.8 | 90.3 | **93.5** | **89.0** | **94.1** | 90.3 |

Table 4: Performance comparison on the DREAMT dataset for Skin Temperature Regression (RMSE) and Sleep Classification (AUROC). FGNO retains competitive performance even when trained with only 5% of labeled data, outperforming the full-data baseline in regression tasks.

| Baseline | 100% labeled data | | 5% labeled data | |
| | RMSE $\downarrow$ | AUROC $\uparrow$ | RMSE $\downarrow$ | AUROC $\uparrow$ |
|---|---|---|---|---|
| BrainBERT/MAE | 0.734 | 0.958 | 0.790 | 0.947 |
| **FGNO (ours)** | **0.600** | **0.965** | **0.710** | **0.954** |

## 4.4 ROBUSTNESS UNDER DATA SCARCITY

Medical data are often costly and limited. To evaluate FGNO's performance in data-scarce scenarios, we designed an experiment where the pre-training phase utilizes most of the available data without labels, after which the downstream probing head was trained on only 5% of the available labeled data and tested on held-out data. As shown in Table 3, FGNO achieves state-of-the-art performance on both SleepEDF and Epilepsy even when only 5% of labeled data is available, outperforming strong baselines. On SleepEDF, our model maintains an accuracy of 93.5% and a macro-F1 of 89.0%, which is nearly identical to the performance obtained with 100% of the labeled data (93.9% ACC, 89.1% MF1). Similarly, on Epilepsy, FGNO achieves 94.1% accuracy and 90.3% MF1 under the 5% setting, matching the results from the full dataset.

We extended this evaluation to the DREAMT dataset. Table 4 demonstrates that even with only 5% of labeled data, FGNO retains competitive performance. For sleep classification, accuracy drops only slightly from 96.5% to 95.4%, and for skin temperature regression, the model maintains an RMSE of 0.71, outperforming the full-data baselines of MAE/BrainBERT (0.734).

These findings highlight the sample efficiency of our approach and suggest that FGNO is particularly well-suited for real-world biomedical applications, where large-scale labeled datasets are often scarce.

## 4.5 ABLATION STUDY AND ANALYSIS

**Clean vs noisy input for probing** A key component of our probing framework involves generating a noisy sample for a given clean input spectrogram $\bar{f}$ and a desired time $s \in [0, 1]$:

$$g_s = s\bar{f} + \sigma_s^{\bar{f}}z, \quad \text{where } z \sim \mathcal{N}(0, I).$$

While effective, this step introduces computational overhead during inference. Moreover, its reliance on a random noise vector $z$ leads to unstable outputs. To investigate a more efficient alternative, we conducted an ablation study where we bypassed noise generation entirely. Instead of feeding the model a noisy input $g_s$, we provided the clean spectrogram $f$ directly and supplied the time value $s$ as an additional conditional embedding (the "Clean Input" method).

Our results, illustrated in Figure 3, show that the Clean Input method yields nearly identical mean performance to the Noisy Input method. For example, at layer 3 and time $s \approx 0.89$, the Clean Input Method achieves a maximum score of 96.40% while the Noisy Input method yields 95.86%. This

confirms the model learns to interpret $s$ as the corruption level. We also found that the Noisy Input method is sensitive to the random noise vector $z$, exhibiting performance variance across runs (std of 0.0039 at the same point). In contrast, the Clean Input method is entirely deterministic. This demonstrates that the clean approach is superior, as it is both more computationally efficient and provides a more stable, reliable inference pathway.

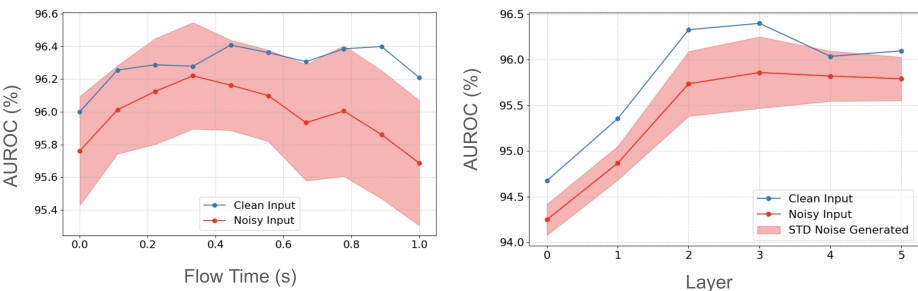

Figure 3: Sleep classification performance (DREAMT dataset, AUROC, %) comparing a "Noisy Input" against a "Clean Input" method across different model layers. **(Left)** For a representative layer, we plot performance as a function of time $s \in [0, 1]$. While both the "Clean Input" and "Noisy Input" methods exhibit the same behavioral trend, the clean input approach yields consistently higher performance. **(Right)** At the optimal time $s \approx 0.89$, the noisy method exhibits high variance over 10 runs (red-shaded region), while the clean method is deterministic and stable.

**Performance across resolutions**   Because FGNO learns the underlying mapping between functions, which are the time-frequency representations via STFT rather than a mapping between fixed-size vectors, it is not inherently constrained to the specific discretization it was trained on. This gives our model the theoretical ability to handle time series at varying resolutions. We tested this property empirically on the BrainTreeBank dataset, which has an original sampling rate of 2048 Hz. We created lower-resolution versions by downsampling the raw 1D time series before applying the STFT. This results in spectrograms with fewer frequency bins; to maintain a consistent input tensor shape for our model, we padded the frequency axis of the downsampled spectrograms with zeros to match the original resolution. We pre-trained a single FGNO model on the original 2048 Hz data and then evaluated its robustness by probing its performance on data across a wide range of lower resolutions, with downsampling factors of 4x, 8x, 12x, 36x, and 48x. To benchmark our function-space approach, we compare our results against MAE and Chronos, a state-of-the-art time-series foundation model that operates directly on the raw 1D data by tokenizing signal values.

As shown in Figure 4, FGNO consistently achieves higher AUROC scores than both baselines across all resolutions, maintaining performance above 74% even under extreme downsampling ($48\times$), whereas MAE drops sharply to around 52% AUROC and Chronos fluctuates around 60%. Unlike FGNO, Chronos operates directly on tokenized raw signal values. To evaluate FGNO on downsampled data, we convert the signal to an STFT and zero-pad the missing high-frequency bins to match the pretrained input dimensionality. Under extreme downsampling, over 90% of the spectrogram contains no signal. Despite this distortion, FGNO maintains a 74%+ AUROC, demonstrating that the learned operator is robust to resolution changes. This result highlights the benefit of learning a resolution-agnostic mapping in function space, enabling FGNO to generalize effectively across sampling rates and outperform strong baselines by a large margin.

## 5 CONCLUSION

In this work, we presented the Flow-Guided Neural Operator (FGNO), a novel self-supervised learning framework that combines flow matching with neural operators for time-series representation learning. By embedding signals via Short-Time Fourier Transform (STFT) into resolution-invariant spectrograms, FGNO preserves multi-scale fidelity without resampling distortions. Our framework pre-trains a single backbone model on a dataset to extract a rich hierarchy of task-specific representations. This is achieved by selecting features from an optimal network layer and a specific flow time, which acts as a continuous control for representation granularity. Furthermore, we demonstrated that

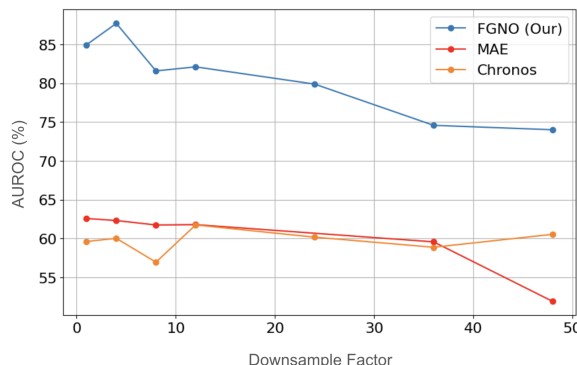

Figure 4: On BrainTreeBank speech classification, our FGNO model, pre-trained once on the original high-resolution data, is evaluated against MAE and Chronos baselines on inputs downsampled by various factors. FGNO consistently outperforms both baselines and shows remarkable stability across resolutions, demonstrating the benefit of learning a resolution-agnostic mapping in function space.

using clean inputs during the probing stage, rather than the noisy inputs common in generative SSL, yields more stable and expressive features. We empirically evaluate FGNO across several biomedical time-series datasets. It demonstrates up to a 35% AUROC improvement in neural signal decoding on BrainTreeBank and a 16% RMSE reduction in skin temperature regression on DREAMT. Critically, FGNO shows exceptional robustness in low-data regimes, maintaining nearly full-data performance on both the SleepEDF and Epilepsy datasets with only 5% of labeled data—an improvement of over 20% against strong competitors. Its effectiveness as a neural operator is confirmed by its stable performance on BrainTreeBank across various downsampling factors, where baselines like MAE and Chronos degrade significantly.

The main limitation of our approach is the reliance on grid search to find the optimal $(l, s)$ pair, though it remains computationally efficient during probing. Future work will aim to automate this selection process, improving efficiency and expanding its applicability to new data modalities. We envision FGNO as a step toward scalable, adaptable SSL, enabling transformative insights from large unlabeled time-series datasets.

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

## A   Implementation Details

**Training details**   The pre-trained model is a 6-layer Transformer designed to process the output of a Short-Time Fourier Transform (STFT). The model's architecture was specifically configured to match the STFT output tensor shape: the model's input dimension of 132 corresponds to the number of frequency bins, and the sequence length of 21 corresponds to the number of time frames. Other key hyperparameters include a hidden dimension of 768, 12 attention heads, a feedforward dimension of 3072, a dropout rate of 0.1, and a learning rate of 0.0001.

**Baselines**   Chronos was implemented using its Hugging Face checkpoint, extracting features with the built-in `encode` function on the same train/test windows as FGNO, though pre-trained on a broader external corpus. MAE baselines followed the BrainBERT implementation. The results of PopT are sourced from its original paper.

**Evaluation metrics**   We evaluated the performance on the two downstream tasks using standard metrics. For the binary sleep classification task (awake vs. asleep), we used the Area Under the Receiver Operating Characteristic curve (AUROC). For the skin temperature regression task, we used Mean Absolute Error, Mean Squared Error (MSE), and Root Mean Squared Error (RMSE) to quantify the model's predictive accuracy.

## B   Data Preprocessing and Splitting

For BrainTreeBank, we adopt the BrainBERT preprocessing pipeline for the raw time series signal data. Afterward, We segment data into non-overlapping 5-second windows, apply STFT (nperseg=400, noverlap=350), and treat each electrode independently. Splitting is done chronologically by subject, the same way it was done in BrainBERT.

For DREAMT, BVP and ACC signals are provided clean. We apply 64 Hz STFT (nperseg=64, noverlap=48) and use 5-second windows.

For SleepEDF/Epilepsy, we use the official TS-TCC preprocessing pipeline to handle data preprocessing. We reuse their released dataloaders to ensure consistency in training. STFT parameters for these two datasets are the same with BrainTreeBank.

Across all datasets, FGNO is pretrained on the training split only. During probing, we use held-out subjects with an $80/10/10$ split for train/validation/test, respectively. Baselines use identical folds.

## C   Computational Efficiency

The table below compares the runtime of FGNO against the MAE/BrainBERT baseline on a single NVIDIA RTX 4090 GPU. Since we finetune only a small probing head instead of the whole model like BrainBERT, our finetuning time is significantly smaller.

Table 5: Runtime comparison between FGNO and baselines across different stages.

| Model | Training | Probing/Finetuning | Inference |
|---|---|---|---|
| FGNO (Ours) | 21h 33m | 2.87 mins | 0.30s |
| BrainBERT/MAE | 19h 53m | 7.17 mins | 0.31s |

## D   Additional Results

Here we provide the detailed performance under different layers and flow times.

Table 6: AUROC (↑) comparison between our model and MAE on DREAMT for sleep classification.

| Layer Number | FGNO (Best AUROC % @ Time) | MAE AUROC % |
|---|---|---|
| 1 | 94.6 @ s=1.00 | **95.8** |
| 2 | 94.6 @ s=0.89 | **95.6** |
| 3 | **96.5** @ s=0.89 | 95.7 |
| 4 | **95.9** @ s=0.67 | 95.4 |
| 5 | **96.2** @ s=0.78 | 95.5 |
| 6 | **96.4** @ s=0.89 | 95.8 |

Table 7: Best RMSE (↓) values against MAE on DREAMT for skin temperature regression task.

| Layer Number | FGNO (Best RMSE °C @ Time) | MAE RMSE °C |
|---|---|---|
| 1 | **0.743** @ s=0.22 | 0.790 |
| 2 | **0.691** @ s=0.33 | 0.775 |
| 3 | **0.656** @ s=0.44 | 0.735 |
| 4 | **0.625** @ s=0.33 | 0.782 |
| 5 | **0.619** @ s=0.44 | 0.738 |
| 6 | **0.600** @ s=0.56 | 0.744 |

Table 8: AUROC (↑) comparison at optimal extraction time on BrainTreeBank for speech detection.

| Layer Number | FGNO (Best AUROC % @ Time) | MAE AUROC % |
|---|---|---|
| 1 | **83.3** @ s=0.778 | 60.7 |
| 2 | **85.8** @ s=0.778 | 67.2 |
| 3 | **86.4** @ s=0.778 | 62.7 |
| 4 | **88.3** @ s=0.889 | 65.5 |
| 5 | **88.6** @ s=0.889 | 63.5 |
| 6 | **88.3** @ s=0.889 | 67.2 |

