# OpenReview forum: "Flow-Guided Neural Operator For Self- Supervised Learning On Time Series Data"
_ICLR.cc/2026/Conference — Submitted to ICLR 2026_

### Official Review · Reviewer_mtcV · 2025-10-29

**Soundness:** 1
**Presentation:** 2
**Contribution:** 2
**Rating:** 2
**Confidence:** 4

**Summary:**

This paper introduces the Flow Guided Neural Operator (FGNO) as a framework for SSL training on 1-dimensional time series.
The paper's goal is to introduce a novel model architecture and measure its performance on downstream classification tasks.

The architecture is simple and elegant: the 1D signal is first vectorized through a short-time Fourier transform and then encoded through a transformer architecture, conditioned sinusoidally with a noise parameter, tailored for each task.

The architecture learns in a self-supervised manner according to a flow-matching objective.
This contrasts with the other existing types of approaches for time-series, based on masked reconstruction (termed masked auto-encoder), next-step predictions, and quantizer approaches (both are referenced through the Chronos paper but not described in the paper). The learning objective and exploration seem interesting and additionally offer a novel inference scheme.

Indeed, an important contribution of this architecture is to allow fine-tuning of the model behavior on independent tasks by varying two hyperparameters: the noise (s) and the layer (l), at which the representations are extracted for classifications.
In opposite, the previous approach could only select a certain layer (l) of representations.

The paper is well structured, original, and clear.

**Strengths:**

Originality:
The paper combines previously known methods into a novel architecture. Notably, they take inspiration from generative SSL methods and adapt them by using clean inputs at test time, solely conditioning on the noise value rather than sampling from noise.

Quality:
The paper explores four different datasets for evaluations.

Clarity:
The architecture is well described.

Significance:
The author makes a significant improvement compared to one other evaluated model (Chronos).

**Weaknesses:**

Major methodological issues:
- It is unclear if the (s,t) parameters are selected using the test data, which would be very problematic, or if the author decomposed the datasets into (train, validation, test) folds. Exact folds and test datasets processing are missing from the methods and appendix.
- It is unclear which data the author used for training the models: similar to test datasets, these dataset folds and preprocessing should be described more extensively.
- Little information is provided on comparative baselines: masked-auto-encoder and Chronos architecture. It is not clear if the authors trained these baselines on the same data as their FGNO model.
- The last section of the paper seems to present conflicting evidence in the figure compared to the text. In the figure, the performance of the FGNO model decreases more than the performance of the Chronos model relative to their baseline value (at a downsampling factor of 1). This does not support the claim "FGNO shows remarkable stability across resolutions".

Moderate methodological issues:
- Two separate claims are made on two non-overlapping sets of datasets.
In principle, it would reinforce the claim to also evaluate how FGNO performs on BrainTree Bank and DREAMT with a change in % of training data; otherwise, we don't know whether the authors decided to use other datasets for this evaluation because the models did better on these datasets in this case. Alternatively, the author should stress why a change in % of training data is better evaluated in these two other datasets.

- Compared to previous papers, the paper is relatively sparse in evaluations.  For example, in the BrainBert paper (Wang et al 2023), which the author extensively cites, the model is evaluated on: Sentence onset, Speech/Nonspeech, Pitch Volume, and Task Avg classification scores. Since this paper touches on a very similar approach, there is no a priori reason to exclude these evaluations.

- Simple baselines. The paper uses two complex baselines: MAE and Chronos. But the quality of these baselines is hard to evaluate without a clear reference to a simpler baseline. For example, previous papers have used Linear Baseline on the STFT features, or simple deep NN on the STFT features (Wang et al 2023).

Minor methodological issues:
- The training details are too short and not very informative.
- No analysis of the model's internal representations is done, such that it remains unclear what the model has exactly learned. Previous papers, for example, in the case of SSL on speech, Baevski et al 2021 have done so. While this could be the scope of future work, the author should touch on whether their architecture allows for similar exploration compared to previous architectures.
- No loss plots of the model training are provided. Although not mandatory, this would help in understanding and replicating the training of the model.

Minor writing issues:
- Line 362: Fig 2 should be Fig 3
- The MAE abbreviation is used several times for different meanings.
- The term "semantic" is used several times throughout the paper, but I would argue that the term "global" better captures the meaning the author would like to convey. Indeed, a very short temporal event could have a semantic relevance (take a sudden seizure event), which is likely captured in the middle-layer; in contrast, a long-lasting change captured by the deeper layer could have little semantic relevance. I would suggest avoiding using "semantic" and focusing on "global" which the author themselves used line 125.

**Questions:**

- How does the FGNO model compare to simple, supervised baselines (Linear, and deep res-net, deep convnets)?

- What does the author mean by "decoder" in their Figure 1?  They seem to evaluate the model on variable prediction (binary or regression), but the variable seems to refer to a reconstruction of the input signal.

- Did the author gain any novel understanding from the biological signal by applying FGNO compared to other models?

- I have not understood why authors refer to the function space operator? The architecture is indeed a transformer model, which does not make it directly act over function spaces but instead acts on the vectorized inputs. My understanding is that this makes the model presentation more complex than it should be and brings little insight into its behavior. I might be wrong here.

- Why did the author select these specific four datasets for evaluation? What are the specific difficulties these dataset offers that the model can deal with, compared to previous datasets?

- Would it be possible for the authors to provide training details (time, amount of data, number of GPU ect...)?

- Would it be possible for the authors to provide details on dataset preprocessing?

---

> ### Author Response · Authors · 2025-11-27
> **Author's Rebuttal (1/3)**
>
> We thank the reviewer for their thoughtful comments and recognizing the novelty, clarity and significance of our flow-matching SSL approach. Below, we will respond to each point in detail and the changes will be reflected in our manuscript.
>
> > [W1] Concern about potential overfitting
>
> We thank the reviewer for raising this important point. We want to note that our dataset is partitioned into distinct train, validation, and test splits. The validation set is kept completely separate from the test set. All model tuning is made exclusively using validation. The performance metrics reported in the paper are from the held-out test set to provide to mitigate the risk of overfitting. Data splitting is chronological and by subject, ensuring no leakage across folds. The information will be included in our Appendix.
>
> > [W2] Clarifying which data are used for training and how preprocessing is performed
>
> We thank the reviewer for raising this important point. We will address data preparation details for each of our datasets.
>
> For BrainTreeBank, we adopt the BrainBERT preprocessing pipeline for the raw time series signal data. Afterward, we segment data into non-overlapping 5-second windows, apply STFT (nperseg=400, noverlap=350), and treat each electrode independently. Splitting is done chronologically by subject, the same way it was done in BrainBERT.
>
> With DREAMT, BVP and ACC signals are provided clean. We apply 64 Hz STFT (nperseg=64, noverlap=48) and use 5-second windows.
>
> For SleepEDF/Epilepsy, we use the official TS-TCC preprocessing pipeline (https://github.com/emadeldeen24/TS-TCC) to handle data preprocessing. We reuse their released dataloaders to ensure consistency in training. STFT parameters for these two datasets are the same with BrainTreeBank.
>
> Across all datasets, FGNO is pretrained on the training split only, with no access to labels. During probing stage, we used held-out subjects and probed only on training/validation subsets. Performance was then collected from the test set, which remains untouched until inference stage. The exact split ratios for all datasets are:
>
> - Pretraining: 90% of the data is used for training and 10% of the data is used for validation
> - Probing: 80% of the data is used for training, 10% of the data is used for validation, and 10% of the data is used for testing
>
> > [W3] Data used for baselines are unclear
>
> We appreciate the reviewer's request for more details on what data was used for baselines. All baselines use the exact same dataset folds as FGNO. For SleepEDF/Epilepsy, we use the preprocessed TS-TCC dataloaders (https://arxiv.org/pdf/2208.06616). For BrainTreeBank, we use the BrainBERT subject-level splits. Chronos uses the same train/test windows extracted from the splits, but is pre-trained on a much wider range of datasets.
>
> > [W4] Multi-resolution analysis is inconsistent
>
> We thank the reviewer for pointing out this observation. We will revise the text to avoid implying that Chronos is more sensitive to resolution.
>
> However, we would like to clarify why FGNO displays a larger absolute drop. Our model is pretrained entirely on high-resolution spectrograms with a fixed number of frequency bins. To evaluate on downsampled data, we follow a simple strategy by converting the downsampled signal to an STFT and zero-padding missing high-frequency bins so that the input dimensionality matches the pretrained model. Under extreme downsampling (e.g., 48×), only a small fraction of the frequency bins contain real signal, while more than 90% of the spectrogram is artificially padded with zeros. Nevertheless, despite this severe distortion, FGNO still maintains >74% AUROC, substantially outperforming Chronos and MAE across all sampling rates.
>
> > [W5] Limited evaluation on BrainTreeBank / DREAMT for the 5% label experiment
>
> We appreciate this suggestion. Following the reviewer’s request, we will incorporate limited-label evaluations for BrainTreeBank and DREAMT. The table below shows the results of the two tasks in the DREAT dataset.
>
> |               | 100% labeled data               | 100% labeled data        | 5% labeled data                 | 5% labeled data          |
> | ------------- | ---------------------------------- | --------------------------- | ---------------------------------- | --------------------------- |
> |               | Skin Temperature Regression | Sleep Classification | Skin Temperature Regression | Sleep Classification |
> | FGNO          | **0.6**                               | **0.965**                      | **0.71**                              | **0.954**                       |
> | BrainBERT/MAE | 0.734                          | 0.958                       | 0.79                             | 0.947                       |
>
> Note that for the skin temperature regression task, the metric is RMSE, so the lower number indicates better result, while for sleep classification, the higher is better.

---

> > ### Author Response · Authors · 2025-11-27
> > **Author's Rebuttal (2/3)**
> >
> > > [W6, W7, Q1] Baselines are simple and limited
> >
> > We thank the reviewer for requesting more baselines and evaluations for our model. In order to expand our baselines, we performed experiments on more tasks for BrainTreeBank and incorporated PopT, a recent (ICLR 2025 Oral Paper) state-of-the-art self-supervised learning model in neural signal decoding. In addition, we will benchmark FGNO against DeepNN model from BrainBERT. We want to note that PopT incorporates explicit domain knowledge about the relationships between intracranial electrodes. Despite not using any such anatomical or multi-channel priors, our model achieves superior performance purely from its flow-matching representation learning, as is shown in the table below.
> >
> > Model comparison in terms of size and accuracy on all four tasks of BrainTreeBank
> > |           | Model size | Speech              | Volume | Pitch | Sentence     |
> > | --------- | ------------ | ------------------- | ------ | ----- | ------------ |
> > | FGNO      | 370K         | **0.925** | **0.919**  | **0.796** | 0.797 |
> > | BrainBERT | 43M          | 0.63                | 0.6    | 0.51  | 0.66         |
> > | PopT      | 20M          | 0.93                | 0.87   | 0.74  | **0.9**         |
> > | Deep NN   | 20M          | 0.7                 | 0.64   | 0.56  | 0.71         |
> >
> >
> > > [W8, Q5] Missing training details
> >
> > We appreciate the reviewer’s request for more training details. We will present the runtime across three stages (training, probing, and inference). Our model was pretrained and probed using 1 RTX 4090 GPU. We hope this will better highlight our model's advantages over other baselines.
> >
> > Runtime comparison
> >
> > |               | Training | Probing/Finetuning | Inference |
> > | ------------- | -------- | ------------------ | --------- |
> > | FGNO          | 21h 33m  | 2.87 mins          | 0.3s      |
> > | BrainBERT/MAE | 19h 53m  | 7.17 mins          | 0.31      |
> >
> >
> > > [W9] Learned representations require more analysis
> >
> > We thank the reviewer for this insightful suggestion and pointing to good references. As the reviewer notes, such explorations is suitable for future work beyond the current scope of our manuscript, which focuses primarily on the methodological contributions and empirical performance of FGNO. That said, our architecture is highly amenable to these analyses: FGNO learns embeddings in a latent vector space, enabling the application of classical high-dimensional visualization techniques (e.g., t-SNE or UMAP). We anticipate these will reveal interpretable trajectories conditioned on noise levels, offering advantages over discrete masking-based SSL methods.
> >
> > > [W10] Missing loss curves
> >
> > We thank the reviewer for requesting more details on the loss curves. We already have and will provide training curves, showing convergence for flow matching pretraining loss and probing head loss.
> >
> > > [W11+] Minor writing issues
> >
> > We would like to thank the reviewer for these careful writing suggestions. We agree with all of them and the changes will be reflected in the draft.
> >
> > > [Q1] Please see [W6]
> >
> > > [Q2] Clarification on “Decoder” in Figure 1
> >
> > In our implementation, the “decoder” in Figure 1 corresponds to a shallow spectrogram reconstruction head used only during pretraining, not during downstream evaluation. This spectrogram prediction head is borrowed from BrainBERT's implementation (https://github.com/czlwang/BrainBERT).
> >
> > > [Q3] New biological insights from FGNO beyond previous models
> >
> > We thank the reviewer for this question. It remains to be explored more, but we have some ideas already. From the FGNO result, we observed the trend that different levels of abstraction require different flow times. We noted that the physiological regression task depends more on global features, which depend on lower to intermediate flow time. In contrast, classification tasks require more local patterns that prefer higher flow time where temporal features are preserved. Therefore, we can use our understanding of the task to determine whether we require higher or lower flow time. Moreover, by using FGNO, we can be exposed to a continuum level of representations from local to global features, allowing us to have more flexibility in performance depending on tasks instead of relying on a single, fixed features from previous models.

---

> > > ### Author Response · Authors · 2025-11-27
> > > **Author's Rebuttal (3/3)**
> > >
> > > > [Q4] Why is FGNO a function space operator?
> > >
> > > We thank the reviewer for raising this point. Although it is true that Transformers operate on vectorized inputs, neural operators always take in discretized vectors, but what makes them operators is the mapping they learn between underlying function spaces, which becomes resolution invariant as the grid is refined.
> > >
> > > In our approach, FGNO operates on spectrograms, which can be viewed as discretized functions over time and frequency space. FGNO’s encoder learns a mapping from this function space to a latent trajectory over flow time. While implemented with a Transformer, the model approximates an operator acting on functions, and can generalize across different discretizations of the same underlying signal. This operator perspective is particularly important in biomedical/health settings, where sampling rates and resolutions vary widely across sensors. Prior SSL time-series work has not examined this property, and our paper is the first to motivate and evaluate it explicitly.
> > >
> > > > [Q5] Please see [W8]
> > >
> > > > [Q6] Clarification on dataset selection
> > >
> > > We appreciate the reviewer’s request for more details regarding our dataset choices. Our goal was to select datasets that provide broad coverage across multiple biomedical domains, including EEG, ECoG, and wearable physiological sensing. These datasets collectively span both classification and regression tasks, allowing us to demonstrate how FGNO’s $(l,s)$ representations adapt across task types. In addition, each dataset contains multiple channels with different sampling rates and resolutions, which is important for evaluating FGNO’s resolution-invariant ability.

---

> > ### Comment · Reviewer_mtcV · 2025-11-27
> > **Request to include the modification in the paper before another round of review**
> >
> > Thank you for these clarifications, notably on cross-validation folds, which was a major soundness point, and which seems to have been well addressed by the authors.
> >
> > Before I take a new deep dive into the article, would it be possible for the author to update the manuscript methods to incorporate the crucial elements indicated in their responses so that the modifications made in the article become explicit?
> > ICLR allows the revision of the manuscript during the rebuttal period.
> >
> > Thank you.

---

### Official Review · Reviewer_Jpnj · 2025-11-04

**Soundness:** 2
**Presentation:** 3
**Contribution:** 3
**Rating:** 4
**Confidence:** 3

**Summary:**

The paper presents a self-supervised method for time-series data based on flow-matching pre-training and short-term Fourier transform (STFT). The method is evaluated on multiple tasks on medical data, with good results at low-label settings.

**Strengths:**

* The motivation is clear.
* The presentation is good and well organized.
* The method is interesting and shows good results in multiple tasks (although not as good as the initial claims).

**Weaknesses:**

* The authors claim that FGNO consistently outperforms established baselines. These comments need to be moderated, as for example Table 2 shows that FGNO performs poorly for Epilepsy 100% labeled data.

* Results from Table 2 with 100% of labeled data are taken from [Emadeldeen et al. 2021] (TS-TCC: https://arxiv.org/abs/2106.14112). This should be clarified explicitly. I would also like to know if results from the 5% part were re-run or copied from another source.

* In Table 2: the performance reduction for FGNO between 100% and 5% labeled data is very small. All other methods show large reductions in performance. This difference is mentioned in the text, but it might merit more analysis or a deeper discussion. It could be related to running the method in different conditions to the baselines (e.g. multiple runs vs single run).

* The paper mentions that the method remains computationally efficient during probing, but there are no details of training and inference times, and no comparison to baselines. Presumably training time of Chronos is very high given the large amounts of data used, but how does inference compare? How does FGNO runtime compares to other baselines?

* There are no details about the baseline models (implementations used, multiple runs or single run, copied results vs re-run, settings, etc.). This is especially true for the for the robustness under data scarcity part (5% of labels in table 2).

* Missing information of STFT parameters (window size W, hop H).

* Making the source code available would be beneficial for reproducibility.

**Questions:**

* Table 2 with 5% data: were these results re-run or copied from another source? Were there multiple runs or a single run?

* How about baselines in other tables? (re-run? multiple runs? settings? implementations used?)

* How does the runtime compare with other methods?

* Please add information about STFT parameters.

---

> ### Author Response · Authors · 2025-11-25
> **Authors' Rebuttal (1/2)**
>
> We thank the reviewer for their thoughtful comments and recognizing the novelty of our approach to use flow-matching pre-training for time-series SSL. Below, we will respond to each point in detail and the changes will be reflected in our manuscript.
>
> > [W1] The claims about FGNO's performance on 100% data should be updated.
>
> We thank the reviewer for pointing out the need to moderate the description in Table 2. The caption for Table 2 will be revised to "... FGNO achieves the strongest performance in the low-label (5%) regime on both datasets and performs competitively under the full-label setting. These results highlight FGNO’s robustness and data efficiency, particularly when labeled data are limited..." We will re-examine related statements to accurately reflect FGNO's performance on Epilepsy.
>
> > [W2] The source of baseline results should be clarified.
>
> We thank the reviewer for pointing out the need for clarification. Regarding Table 2, all baseline numbers (both 100% and 5% labeled) are copied directly from Table 2 in Emadeldeen et al. (https://arxiv.org/pdf/2208.06616). Note this is a later version of the link that the reviewer suggested. We will reflect this information in experiment setups.
>
> > [W3] Data efficiency needs deeper discussion.
>
> We agree that FGNO’s little drop in performance between 100% and 5% labeled data merits deeper discussion. The core reason behind FGNO’s robustness is inherent to the flow matching pretraining objective. We can interpret this as a form of “self-augmentation.” For example, MAE relies on a single, fixed corruption process (a 75% mask ratio). Additionally, contrastive methods (CPC, TS-TCC) learn only from a small, discrete set of augmentations applied at the instance level. Both of these approaches explore a narrow region of the data manifold. However, FGNO’s flow matching pretraining objective is fundamentally generative. By training the model to denoise inputs sampled from the entire flow trajectory, the encoder effectively learns from a continuum of noise levels. This exposes the encoder to a virtually unbounded set of perturbations, leading to a much more structured and complete latent representation. Consequently, when only 5% of labeled data is available, the probing head operates on a better latent space. In contrast, baseline encoders trained from weaker or more limited pre-training signals do not form such a robust representation, which explains their degraded performance in the scarcity setting.
>
> > [W4] Lack of runtime report
>
> We appreciate the reviewer’s request for runtime clarification. We will present the runtime across three stages (training, probing, and inference) along with new comparisons on model size and accuracy using three additional tasks and a new baseline (PopT, ICLR 2025 Oral). We hope this will better highlight our model's advantages over other baselines.
>
> Runtime comparison
>
> |               | Training | Probing/Finetuning | Inference |
> | ------------- | -------- | ------------------ | --------- |
> | FGNO          | 21h 33m  | 2.87 mins          | 0.3s      |
> | BrainBERT/MAE | 19h 53m  | 7.17 mins          | 0.31      |
>
> Model comparison in terms of size and accuracy on all four tasks of BrainTreeBank
>
> |           | Model size | Speech              | Volume | Pitch | Sentence     |
> | --------- | ------------ | ------------------- | ------ | ----- | ------------ |
> | FGNO      | 370K         | **0.925** | **0.919**  | **0.796** | 0.797 |
> | BrainBERT | 43M          | 0.63                | 0.6    | 0.51  | 0.66         |
> | PopT      | 20M          | 0.93                | 0.87   | 0.74  | **0.9**         |
> | Deep NN   | 20M          | 0.7                 | 0.64   | 0.56  | 0.71         |
>
> > [W5] Experimental setup details
>
> We thank the reviewer for highlighting the need for more details in our experimental setup. For the Epilepsy and Sleep-EDF datasets, we adopted the exact preprocessing pipeline from TS-TCC [Emadeldeen et al., 2021] including signal normalization and segmentation. We then train our model on the resulting dataloaders that ensure identical input distributions. For Chronos, we adopt it from huggingface and extract its feature using the built-in `encode` function. All of our experiments are single runs that are randomly initialized.

---

> ### Author Response · Authors · 2025-11-25
> **Authors' Rebuttal (2/2)**
>
> > [W6] STFT parameters are missing
>
> We thank the reviewer for requesting data preprocessing parameters. For BrainTreeBank, SleepEDF, and Epilepsy, we use nperseg = 400 and noverlap = 350, which correspond to 400-sample windows with 87.5% overlap. For DREAMT’s BVP signal (64 Hz sampling rate), we use nperseg = 64 and noverlap = 48 to accommodate the lower sampling frequency. Across all datasets, the continuous recordings are segmented into non-overlapping 5-second windows, and each 5-second window is independently transformed into a spectrogram and fed into the model during pretraining and probing.
>
> > [W7] the availability of source code
>
> We thank the reviewer for their interest in the source code. We are committed to the open-source community and will release the full reproducible code upon acceptance.
>
> The questions are reflected separately in the answers above.

---

### Official Review · Reviewer_YGy1 · 2025-11-06

**Soundness:** 2
**Presentation:** 3
**Contribution:** 2
**Rating:** 4
**Confidence:** 3

**Summary:**

The paper introduces a self-supervised pretraining approach for time series data that relies on a flow matching objective. Univariate time series segments are represented using the short-time Fourier transform, and fed to a Transformer along with a denoising schedule parameter, used as conditioning. The model is then trained to predict the velocity of the denoising process at given schedule values. At inference time, both the schedule dimension and the model depth can be varied to find the optimal coarseness for a given downstream task. This approach is evaluated on biomedical downstream tasks based on intracortical and surface EEG, blood volume pulse and accelerometers time series. Results suggest the proposed approach outperforms existing baselines, including a time series foundation model, across the four tasks and in low labeled data regimes.

**Strengths:**

Originality:
* The combination of a flow matching objective for time series representation learning, along with the use of an STFT input representation, appears novel.

Clarity:
* The paper is well written and easy to follow despite some important methodological information missing.

Significance
* The results suggest that NFGO outperforms strong time series foundation model baselines. This kind of approach could be an interesting alternative to masked autoencoding and autoregressive approaches that are commonly applied on time series data to pretrain foundation models.

**Weaknesses:**

1. The lack of variability estimates makes it hard to compare performance with the baseline models. For instance, the three models reported in Table 1 are very close to each other on sleep classification.
2. There are multiple missing critical methodological details, notably about datasets preparation and splitting (Q1-4).
3. The set of reported baseline models is limited, especially in Table 1. It is unclear why different baselines were used in Tables 1 and 2. There is also missing information about how the different baselines were pretrained/trained and evaluated, for instance MAE, CPC, SimCLR and the supervised baselines. Finally, especially given there are only four datasets, it would be informative to also report for each task the results obtained by domain/dataset-specific approaches as a point of reference (as is already done for the DREAMT dataset in Section 4.2).

**Questions:**

1. If I understand correctly, each NGFO model is pretrained on the downstream dataset it is going to be evaluated with, without access to labels. It is then reused in a supervised linear probing fashion using the labels. How are the datasets split for pretraining, supervised training, model selection, and final evaluation?  In the case of multivariate datasets, were all channels considered separately, and was the splitting done chronologically? Are the baselines (e.g. MAE, CPC, SimCLR, etc.) also pretrained then linear probed/finetuned using the same scheme? Does Chronos solely use existing external model checkpoints?
2. How were the different datasets preprocessed (assuming other steps were involved to get from the raw data to the STFT)? For instance, the windowing strategy will dramatically impact the number of examples available for training and can introduce leakage if done improperly.
3. Are the final performances reported for NGFO in e.g. Tables 1 and 2 obtained on the same set used for selecting the optimal depth and time, i.e. what is the difference between the validation set of Equation 7 and the final evaluation set? If so, these results are likely to be overfitted as compared to those of the baselines. To make baselines more comparable, a similar layer-wise evaluation of representations could be carried out for Chronos and the other baseline models (i.e. is another layer’s representation better than the last one?).
4. How were the 5% of labels selected for the experiment of Table 2? On biomedical tasks, a realistic scenario would be to keep 5% of subjects (or recordings) rather than randomly sample 5% of the training examples uniformly. If the latter strategy was used, this could explain the relatively similar performance obtained by the different methods on such a small percentage of the original training examples.
5. Analysis of Figure 4: while FGNO achieves higher performance across the range of tested downsampling factors, the baselines actually do appear more stable; for instance, Chronos is consistently around 60% AUROC across all factors. FGNO, on the other hand, saw a decrease of around 10%. This contradicts the following claim: “Unlike FGNO, Chronos’s performance is more directly tied to the sampling rate of the input sequence.” (also repeated at line 480). I’d argue this is not true based on this figure, and if anything, this analysis suggests that while FGNO performs better than the baselines, it is more affected by changing input sampling frequencies than e.g. Chronos. Reporting variability would also help clarify this trend.
6. How does the proposed method compare to the baselines in terms of computational complexity?
7. How were the hyperparameters of the transformer selected, and does the approach scale with the quantity of training data/the number of parameters? Results on skin temperature regression (Figure 2) suggest that the representations of deeper layers are better for this downstream task, suggesting that deeper models might be beneficial.

Notes:
* The abstract says the model is evaluated across three biomedical domains, but I believe this should be four.

---

> ### Author Response · Authors · 2025-11-26
> **Authors' Rebuttal (1/2)**
>
> We thank the reviewer for their thoughtful comments and recognizing the novelty of our approach to use flow-matching pre-training for time-series SSL. Below, we will respond to each point in detail and the changes will be reflected in our manuscript.
>
> > [W1] Lack of variability estimate
>
> We thank the reviewer for highlighting the importance of variability estimates. For the "noisy method", each performance was collected by averaging the results from 10 sample runs with different random seeds. To further address this concern, we will re-run FGNO experiments for the "clean method" using multiple random seeds, and we will report mean ± standard deviation for all key metrics in Table 1 and Table 2.
>
> > [W2] Missing methodological details for data handling
>
> We appreciate the reviewer’s request for more experimental clarity. We will address data preparation details for each of our datasets.
>
> For BrainTreeBank, we adopt the BrainBERT preprocessing pipeline for the raw time series signal data. Afterward, we segment data into non-overlapping 5-second windows, apply STFT (nperseg=400, noverlap=350), and treat each electrode independently. Splitting is done chronologically by subject, the same way it was done in BrainBERT.
>
> With DREAMT, BVP and ACC signals are provided clean. We apply 64 Hz STFT (nperseg=64, noverlap=48) and use 5-second windows.
>
> For SleepEDF/Epilepsy, we use the official TS-TCC preprocessing pipeline (https://github.com/emadeldeen24/TS-TCC) to handle data preprocessing. We reuse their released dataloaders to ensure fairness in training. STFT parameters for these two datasets are the same with BrainTreeBank.
>
> Across all datasets, FGNO is pretrained on the training split only, with no access to labels. During probing stage, we used held-out subjects and probed only on training/validation subsets. Performance was then collected from the test set, which remains untouched until the inference stage.
>
> > [W3] Limited and inconsistent baselines
>
> We thank the reviewer for requesting details of our baselines. The reason for the differing baselines is that each dataset uses different benchmark suites:
>
> - Table 1 (BrainTreeBank & DREAMT) follows the baselines used in BrainBERT and the DREAMT paper, which used boosted tree models.
>
> - Table 2 (SleepEDF & Epilepsy) follows the SSL classification baselines of TS-TCC.
>
> In order to expand our baselines, we performed experiments on more tasks for BrainTreeBank and incorporated PopT, a recent (ICLR 2025 Oral Paper) state-of-the-art self-supervised learning model in neural signal decoding. In addition, we will benchmark FGNO against DeepNN model from BrainBERT. We want to note that PopT leverages BrainBERT embeddings as its foundational representation and incorporates explicit domain knowledge about the relationships between intracranial electrodes, reflecting the fact that different brain regions contribute distinct information. Despite not using any such anatomical or multi-channel priors, our model achieves superior performance solely through its flow-matching representation learning, as shown in the table below.
>
> Model comparison in terms of size and accuracy on all four tasks of BrainTreeBank
>
> |           | Model size | Speech              | Volume | Pitch | Sentence     |
> | --------- | ------------ | ------------------- | ------ | ----- | ------------ |
> | FGNO      | 370K         | **0.925** | **0.919**  | **0.796** | 0.797 |
> | BrainBERT | 43M          | 0.63                | 0.6    | 0.51  | 0.66         |
> | PopT      | 20M          | 0.93                | 0.87   | 0.74  | **0.9**         |
> | Deep NN   | 20M          | 0.7                 | 0.64   | 0.56  | 0.71         |
>
> > [W4] Missing details on baseline training/evaluation
>
> We thank the reviewer for pointing out the need for clarification. Regarding Table 2, all baseline numbers (both 100% and 5% labeled) are copied from Table 2 in Emadeldeen et al. (https://arxiv.org/pdf/2208.06616). Note this is a later version of the link that the reviewer suggested. We will reflect this information in experiment setups. For Chronos, we adopt it from huggingface and extract its feature using the built-in `encode` function. All of our experiments are single runs that are randomly initialized. For BrainBERT, PopT, and DeepNN, the performance reported was copied from the original papers (BrainBERT: https://arxiv.org/abs/2302.14367, PopT: https://arxiv.org/abs/2406.03044)

---

> ### Author Response · Authors · 2025-11-26
> **Authors' Rebuttal (2/2)**
>
> > [W5] Potential overfitting and representation selection
>
> We thank the reviewer for raising this important point. We emphasize that the validation set used to select the optimal (l, s) pair is strictly separate from the held-out test set. This ensures that the representation selection step does not leak information into the test evaluation.
>
> Regarding baselines, Chronos does not undergo layer-wise search because its architecture is different from FGNO. Chronos uses a single encoder module according to the official HuggingFace `encode` function. The model does not expose intermediate layers designed for our purpose of feature extraction. Therefore, our use of Chronos embeddings is consistent with standard practice.
>
> > [W6] Strategy for selecting 5% labels
>
> We thank the reviewer for this careful thought. For this experiment, we adopted the exact preprocessing pipeline from TS-TCC [Emadeldeen et al., 2021] including signal normalization and segmentation. We agree that subject-level 5% sampling is more realistic for clinical settings, but we want to follow TS-TCC and sample 5% of training examples uniformly to ensure fairness in our experiment. Additionally, we want to highlight that the extracted dataloader was split into train/validation/test set, allowing the reported performance to be unbiased and accurate.
>
> > [W7] Claim on the robustness of FGNO on multi-resolution should be updated
>
> We thank the reviewer for this careful observation. We will revise the text to avoid implying that Chronos is more sensitive to resolution.
>
> However, we would like to clarify why FGNO displays a larger absolute drop. Our model is pretrained entirely on high resolution spectrograms with a fixed number of frequency bins. To evaluate on downsampled data, we follow a simple strategy by converting the downsampled signal to an STFT and zero-padding missing high-frequency bins so that the input dimensionality matches the pretrained model. Under extreme downsampling (e.g., 48×), only a small fraction of the frequency bins contain real signal, while more than 90% of the spectrogram is artificially padded with zeros. Nevertheless, despite this severe distortion, FGNO still maintains >74% AUROC, substantially outperforming Chronos and MAE across all sampling rates.
>
> Following the reviewer's suggestion in another reply, we will add variability estimates (mean ± std) across runs to clarify the stability trend.
>
> > [W8] Computational complexity comparison
>
> We appreciate the reviewer’s request for runtime clarification. We will present the runtime across three stages (training, probing, and inference). We hope this will better highlight our model's advantages over other baselines.
>
> Runtime comparison
>
> |               | Training | Probing/Finetuning | Inference |
> | ------------- | -------- | ------------------ | --------- |
> | FGNO          | 21h 33m  | 2.87 mins          | 0.3s      |
> | BrainBERT/MAE | 19h 53m  | 7.17 mins          | 0.31      |
>
> > [W9] Transformer hyperparameter selection and scalability
>
> We thank the reviewer for the valuable question. Hyperparameters were selected based on BrainBERT model (https://github.com/czlwang/BrainBERT) to ensure fair comparison. Our FGNO model scales well because it performs competitively with its baselines while being a much smaller model as shown in the table [W3] on model accuracy comparison in terms of size. We agree that future work may explore further on scalability of such SSL methods.

---

### Official Review · Reviewer_6M52 · 2025-11-08

**Soundness:** 3
**Presentation:** 3
**Contribution:** 2
**Rating:** 2
**Confidence:** 4

**Summary:**

The paper proposes an SSL approach for time series modelling. Input is first mapped into a short-time fourier space and varying strengths of noise are then applied at different network layers and flow times. The model is then pre-trained in a self-supervised fashion using a flow matching objective. Once pre-trained, representation at one of the layers is chosen for a supervised objective trained for the target task with the rest of the backbone frozen.

**Strengths:**

The paper is well written and easy to follow. Authors propose an interesting approach to use adaptive noise and representation layer for SSL. Empirical results on real-world datasets show that the proposed approach can outperform strong baselines. Authors also demonstrate that different settings for layer and noise level are needed to achieve top performance across datasets.

**Weaknesses:**

I have major concerns with the experiments section. First, authors use vary outdated baselines. In Table 2 the most recent baseline is almost four years old. Numerous time series methods have been proposed recently both foundation and task specific. GIFT-Eval is a popular benchmark for these methods (https://huggingface.co/spaces/Salesforce/GIFT-Eval) and many recent leading methods are listed there. I think a thorough comparison against leading recent baselines is needed to confirm the performance of the proposed method. Second, there is very little analysis of complexity, runtime or other aspects of FGNO. Equation 7 can be computationally expensive as one would have to sweep over many pairwise settings especially for deep models with many layers. There is also a risk of overfitting if validation set if probed repeatedly to select a fine grained setting. However, authors do not provide any runtime analysis or strategies to mitigate this complexity. I believe that the experimental section needs a major revision before this paper can be accepted for publication.

**Questions:**

Do you have runtime complexity to select optimal (l, s) pair in equation 7? Also, how stable is that hyper parameter landscape?

---

> ### Author Response · Authors · 2025-11-25
> **Authors' Rebuttal (1/2)**
>
> We thank the reviewer for their thoughtful comments and recognizing the novelty of our approach to use adaptive noise and layer-wise representation for SSL. Below, we will respond to each point in detail and the changes will be reflected in our manuscript.
>
> > [W1.1] Baselines are outdated
>
> Thank you for your suggestion. In order to make our baseline more updated, we will incorporate PopT, a recent (ICLR 2025 Oral Paper) state-of-the-art self-supervised learning model in neural signal decoding. PopT leverages BrainBERT embeddings as its foundational representation and incorporates explicit domain knowledge about the relationships between intracranial electrodes, reflecting the fact that different brain regions contribute distinct information. Despite not using any such anatomical or multi-channel priors, our model achieves superior performance purely from its flow-matching representation learning, as is shown in the table below.
>
> Model comparison in terms of size and accuracy on all four tasks of BrainTreeBank
>
> |           | Model size | Speech              | Volume | Pitch | Sentence     |
> | --------- | ------------ | ------------------- | ------ | ----- | ------------ |
> | FGNO (Ours)      | 370K         | **0.925** | **0.919**  | **0.796** | 0.797 |
> | BrainBERT | 43M          | 0.63                | 0.6    | 0.51  | 0.66         |
> | PopT      | 20M          | 0.93                | 0.87   | 0.74  | **0.9**         |
> | Deep NN   | 20M          | 0.7                 | 0.64   | 0.56  | 0.71         |
>
> We believe including PopT and 3 new tasks (Volume/Pitch/Sentence classification) will strengthen our evaluation. Following your suggestion, we will evaluate up-to-date foundation models for Table 2, including TimesFM (Google), MOIRAI (Salesforce), and FlowState (IBM).
>
> > [W1.2] GIFT-Eval is suggested
>
> Thank you for highlighting the importance of evaluating FGNO against recent time-series models. We appreciate the pointer to GIFT-Eval. However, we would like to clarify that our experiments are fundamentally representation-learning and probing tasks:
>
> - BrainTreeBank (neural signal decoding)
> - DREAMT (classification + regression)
> - SleepEDF (5-class sleep-stage classification)
> - Epilepsy (binary EEG seizure detection)
>
> These tasks evaluate discriminative features extracted via probing, not forecasting performance. In contrast, the models featured in GIFT-Eval are designed primarily around forecasting, often trained with decoder architectures and probabilistic forecasting losses. These models assume autoregressive tokenization or predictive causal structure which are not compatible with the BERT-style masked or FGNO framework we use. Our method, FGNO, operates on clean or noisy spectrogram inputs, learns noise conditioned internal representations, and is evaluated using probing heads, which is a completely different set up compared to forecasting models. Thus, while GIFT-Eval is highly relevant for forecasting foundation models, it does not measure the intended capability of FGNO. Additionally, forecasting is rarely the primary benchmark in healthcare. Instead, representation quality for classification/regression is far more relevant. Thus, we respectfully note that they are not architecturally aligned with our pretraining or task structure.
>
> > [W2.1] Analysis of complexity or runtime of FGNO
>
> To address your concern on runtime and $ (l, s) $ selection, we will provide the following details in the paper: Equation 7 requires a grid search over 6 layers × 10 flow times ≈ 60 configurations. Each configuration trains only a lightweight linear probing head (backbone frozen), taking 2–5 minutes per configuration on a single GPU. The total cost is 2–4 GPU-hours per dataset. This is substantially cheaper than masked autoencoding counterpart like BrainBERT that requires the model to be fully finetuned (with weight unfrozen) as well as additional linear layers over top. Additionally, FGNO provides an additional degree of freedom, flow time, that offers substantial performance gains. For more information, please refer to the model size vs accuracy table in [W1.1] as well as the runtime comparison table for the DREAMT dataset below.
>
> |               | Training | Probing/Finetuning | Inference |
> | ------------- | -------- | ------------------ | --------- |
> | FGNO          | 21h 33m  | 2.87 mins          | 0.30s      |
> | BrainBERT/MAE | 19h 53m  | 7.17 mins          | 0.31s      |
>
> > [W2.2] overfitting concern
>
> Regarding to your concern that our method repeated probes validation set to find the most optimal $(l, s)$ pair, we want to note that our dataset is partitioned into distinct train, validation, and test splits. The validation set is kept completely separate from the test set. All model tuning is made exclusively using validation. The performance metrics reported in the paper are from the held-out test set to provide to mitigate the risk of overfitting.

---

> ### Author Response · Authors · 2025-11-25
> **Authors' Rebuttal (2/2)**
>
> > [Q1] How stable is (l, s)'s landscape?
>
> Regarding the stability of the hyperparameter landscape for selecting the optimal $(l, s)$ pair, we have visualized this in Figure 2 and Figure 3 of the manuscript, which shows a relatively smooth loss landscape across different $l$ and $s$ values. Moreover, our clean-input probing strategy during evaluation eliminates stochasticity in the representations (as the flow matching objective is deterministic given fixed noise levels), yielding fully reproducible results without reliance on random sampling. This further enhances the stability of the landscape. We will add a brief discussion of these properties in Section 4.5 to address this concern.

---

### Meta-Review · Area_Chair_ChrC · 2026-01-08

**Summary:**

This paper introduces the Flow-Guided Neural Operator (FGNO), a self-supervised framework for time-series representation learning that combines flow matching with STFT-based inputs and leverages noise level and network depth to extract multi-scale features. Experiments on biomedical datasets indicate strong performance, particularly in low-label settings.

Reviewers found the approach original and clearly presented but raised concerns about experimental completeness, baseline coverage, methodological clarity, and the strength of some claims. The rebuttal addressed many of these issues through expanded experiments, updated baselines, clarified data handling, and added runtime analyses, substantially improving the manuscript. Nonetheless, questions remain regarding the breadth of comparisons, the largely empirical nature of the contribution, and the limited analysis of learned representations. With further consolidation and clearer conceptual positioning, the work could become a strong future submission.

**Reviewer Concerns:**

The rebuttal addressed concerns regarding baseline coverage, experimental clarity, data splits, runtime and model size reporting, and overly strong claims, while questions about the breadth of comparisons, deeper analysis of learned representations, and the strength of the neural operator framing remain open.

**Reviewer Scores:**

Reviewer assessments initially ranged from clear rejection to marginally below the acceptance threshold. Following the rebuttal, several evaluations would likely shift upward in light of the additional experiments and clarifications, although the manuscript would still be positioned near the acceptance boundary.

---

### Decision · Program_Chairs · 2026-01-26

Reject